

# Research progress on the bulb expansion and starch enrichment in taro *(Colocasia esculenta (L). Schott)*

Erjin Zhang, Wenyuan Shen, Weijie Jiang, Wenlong Li, Xiaping Wan, Xurun Yu and Fei Xiong

Yangzhou University, Yangzhou, Jiangsu Province, China

## ABSTRACT

**Background**. Taro is an important potato crop, which can be used as food, vegetable, feed, and industrial raw material. The yield and quality of taro are primarily determined by the expansion degree of taro bulb and the filling condition of starch, whereas the expansion of taro bulb is a complex biological process. However, little information is reviewed on the research progress of bulb expansion and starch enrichment in taro.

**Methodology**. PubMed, Web of Science, and the China National Knowledge Infrastructure databases were searched for relevant articles. After removing duplicate articles and articles with little relevance, 73 articles were selected for review.

**Results**. This article introduces the formation and development of taro bulb for workers engaged in taro research. The content includes the process of amyloplast formation at the cytological level and changes in bulb expansion and starch enrichment at physiological levels, which involve endogenous hormones and key enzyme genes for starch synthesis. The effects of environment and cultivation methods on taro bulb expansion were also reviewed.

**Conclusions**. Future research directions and research focus about the development of taro bulb were proposed. Limited research has been conducted on the physiological mechanism and hormone regulatory pathway of taro growth and development, taro bulb expansion, key gene expression, and starch enrichment. Therefore, the abovementioned research will become the key research direction in the future.

## INTRODUCTION

Taro is an underground bulbous crop planted in the tropical and subtropical regions. It is originated in China, India, Malaysia, and other regions and is widely cultivated in Asia, Africa, and other regions. It has a cultivation history of more than 2,000 years in China. At present, taro can be classified using three methods. Based on the ecological type, taro can be classified into two, namely, aquatic taro planted in paddy field and dry taro planted in orchard soil. Based on the eating part, taro can be classified into three, namely, petiole taro whose main edible part is the petiole, flower taro whose main edible part is the flower, and bulb taro whose main edible part is the bulb. Based on the bulb tillering habit, taro can be classified into three, namely, Kui taro, multi-cormel taro, and multi-head taro

Corresponding author
Fei Xiong, feixiong@yzu.edu.cn

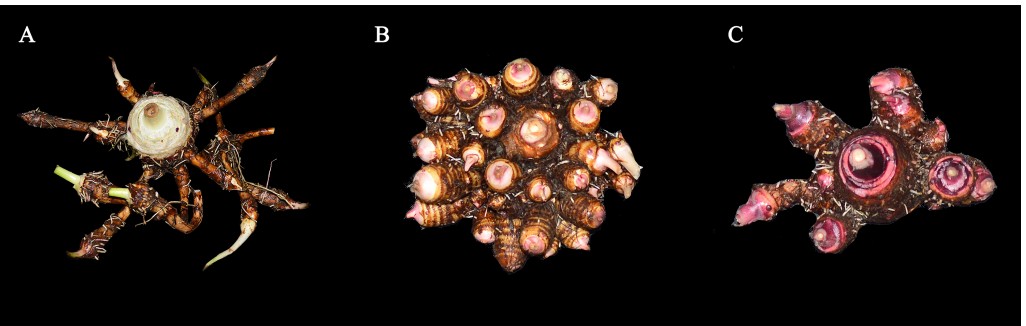

**Figure 1 Three classifications of taros in agricultural production of China.** (A) Kui taro; (B) multi-cormels taro; (C) multi-head taro. Zhang identified the three samples in 1982 (*Zhang, 1982*).

(*Zhang, 1982*; *Wu, Chang & Wang, 2021*). The three classifications of taro in agricultural production of China exhibit various characteristics (Fig. 1). Kui taro has few sub taro and single mother taro, which is the main edible part. In multi-cormel taro, several sub taros grow in groups, which have a strong tillering ability. In multi-head taro, few sub taros and tillers of mother taros grow in groups. The fundamental difference of taro bulb tiller is the expansion between mother and sub taro. This phenomenon is due to the difference in the number of chromosome and differential expression of genes (*Zhu et al., 2018*). Studies on taro chromosomes have proven that Kui taro is diploid ($2n = 2x = 28$), whereas multi-head and multi-cormel taro are triploid ($2n = 3x = 42$) (*Huang et al., 2014*; *Wu, Chang & Wang, 2021*). Chromosome multiples are related to the geographical distribution. Diploid is more common in hot and humid areas with low altitudes, and triploid is more common in dry and cold areas with high altitudes (*Zhang & Zhang, 1990*).

The taro bulb, which is rich in starch and carbohydrates, is a staple food (*Njintang, Scher & Mbofung, 2008*). Compared with potato, sweet potato, and cassava, taro starch granule is small, and it has a diameter of approximately 1.5 $\mu$m, making it easy to digest and providing therapeutic and health care functions. Taro starch granule also has good cold and hot stability, whiteness, and adhesion. Therefore, it can be used as a brightener in cosmetics (*Qi, Yin & Zheng, 2015*). Taro bulbs are rich in many nutrients, such as starch, and the starch content of different varieties of taro could reach 10%–36% (*Falade & Okafor, 2013*). Taro starch can be used as an industrial raw material. It could improve the stability of Pickering emulsion (*Zhang et al., 2020*), and it could be used for the treatment of wastewater generated during printing and dyeing (*Zhou et al., 2018*) and as an enhancer of starch film (*Dai et al., 2015*).

Taro contains a large amount of dietary fiber and various essential amino acids, but its fat content is low (*Sefa-Dedeh & Agyir-Sackey, 2002*). It is also rich in vitamins and amino acids, making it a food for people of all ages (*Han et al., 2018*). Taro leaf contains flavonoids, but biosynthesis of flavonoids in taro bulb remains unknown (*Iwashina et al., 1999*). Taro is a food and medicine homologous crop, which is often used for the treatment of diarrhea, internal bleeding, asthma, and skin diseases (*Prajapati et al., 2011*). It could

also lower blood sugar and cholesterol levels (*Sebnem & Sedef, 2012*). Therefore, taro has good application potential for medicinal development.

Based on the statistics of the Food and Agriculture Organization of United Nations, taro is the 14th largest vegetable crop, with a global planting area of 1.6 million hectares and an annual output of 11.7 million tons. The cultivation area in China ranks first in the world, and it has a large number of wild resources and local varieties, which are primarily distributed in southern regions such as Yunnan, Taiwan, and the Yangtze River Basin. In recent years, taro has become an important input and export trade product. The total export trade volume and total value were higher than the imports (*Chang & Wang, 2019*). With the rapid development of the taro industry, the demand for taro has increased. It has also introduced new and high requirements for taro production and basic theoretical research.

## SURVEY METHODOLOGY

PubMed, Web of Science, and the China National Knowledge Infrastructure databases were searched for relevant articles. A total of 2,525 articles appeared in the PubMed database using "taro" as the search term, and the date of publication was from 1975/1/1 to 2022/5/1. After narrowing the search using the keywords "bulb of taro," "bulb expansion in taro," "bulb starch enrichment in taro," "amyloplast enrichment process of taro bulbs," "development of taro bulbs," "regulation of hormones on bulb development," "genes in starch synthesis," and "regulation of hormones on bulb swelling," 1,532 studies were obtained. Using "taro" as the search term, 2,525 articles published between 1975 and 2022 appeared in the Web of Science database, of which 914 articles were selected as previously described. After removing duplicate articles and articles with little relevance, 73 articles were selected for review.

### Taro expansion and developmental process

Taro is a perennial herbaceous plant, which belongs to the Araceae family, but it is generally cultivated as an annual crop in agricultural production. *Ivancic & Lebot (1999)* identified this plant as *Colocasia esculenta* based on its morphological examination. Based on the growth and developmental characteristic of taro (Fig. 2), its life circle is divided into five periods, namely, embryonic stage, seedling stage, plant vigorous growth stage, taro expansion stage, and bulb dormancy stage (*Sun, Sun & Yuan, 2014*). Taro bulbs undergo metamorphosis during evolution. According to an authoritative book (*Huang & Ke, 2006*), some terms such as "mother taro," "grandson taro," and "great grandson taro" are direct translation of local Chinese terms. Here, multi-cormel taro and multi-head taro are used as examples. The bulb of taro is composed of mother and sub taro, and some varieties include grandson and great grandson taro (Fig. 3). The mother taro of taro bulb is developed by the top bud of the seed taro. The sub taro is developed from the lateral bud of the mother taro, and the grandson taro is developed from the lateral bud of the sub taro. An axillary bud is present on each node of mother taro, and it can develop into sub taro. The axillary bud grown in the leaf axil and the leaf have a cogrowth relationship. The growth site grows clockwise, and the angle between adjacent axillary buds is 144°. The position of

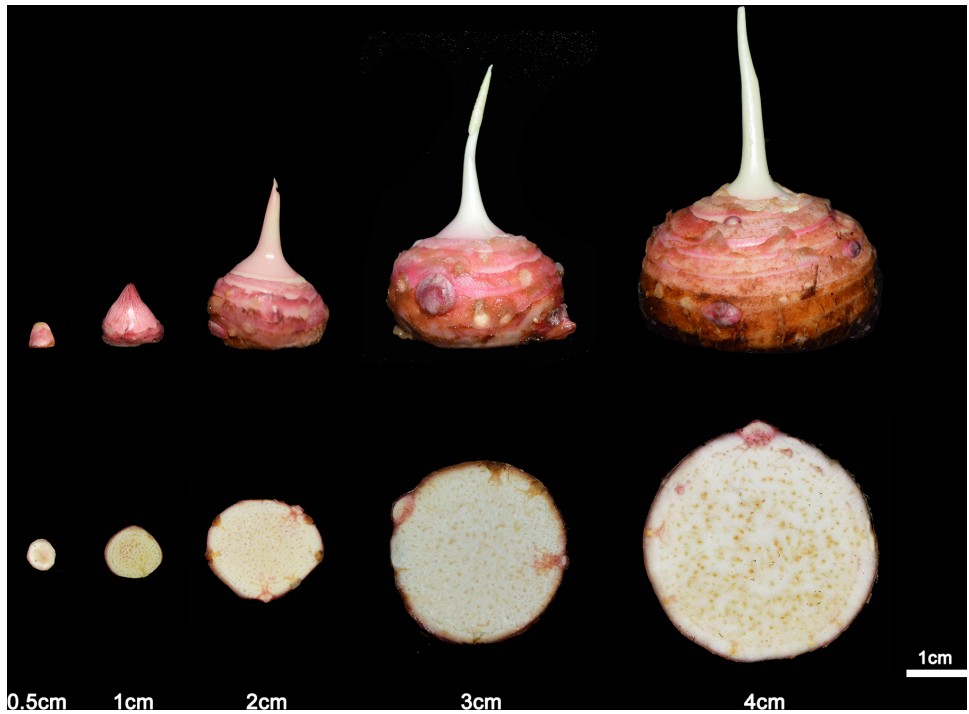

**Figure 2** **The development of taro according to the diameter.** The taro bulb is developed by the bud and the diameter increases with the development process. From left to right, the age is 2 weeks, 4 weeks, 8 weeks, 14 weeks and 16 weeks.

each axillary bud is roughly in a straight line, showing typical 2/5 phyllodes features (*Xu et al., 2020*). The sub taro of Kui taro initially grows and then thickens, and the top swells to form a smaller sub taro. The sub taro of taro with multiple cormels elongates and thickens, and the whole bulb also swells to form a conical shape. Three small-molecular-weight specific proteins during bulb development in the sub taro, namely, CSP2, CSP3, and CSP4, have spatiotemporal specificity, and they are related to the occurrence and expansion of the sub taro. The relative difference in the content of CSP1, CSP2, and CSP4 in the mother taro is related to the apical dominance of the terminal buds of the mother taro. The greater the difference is, the more evident the apical dominance and the lower the developmental degree of the sub taro (*Zhu et al., 2018*). However, *Castro et al. (1992)* found that the expression of the gene (TC1), which encodes a corm globulin protein (G1d) related to curculin, is a spatiotemporal globulin protein during bulb development. The gene that determines bulb expansion is present at the beginning of bulb differentiation. However, the role of the TC1 gene in bulb expansion remains to be further studied. The proportion of fresh weight of taro initially increases and then decreases during growth. Similarly, the water content of the bulb initially increases and then decreases, whereas the dry matter content gradually increases, showing an S-shaped growth trend. The dry matter accumulation of the sub taro in the later stage is greater than that of the mother taro (*Xu et al., 2020*).

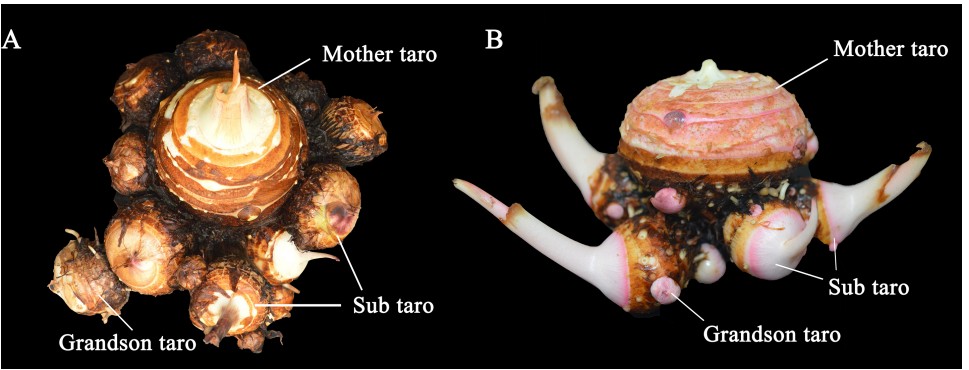

**Figure 3** **The taro bulb's mother taro, sub taro and grandson taro.** (A) Multi-head taro; (B) multi-cormels taro. Mother taro: it is developed by the top bud of the seed taro. Sub taro: it is developed from the lateral bud on the mother taro. Grandson taro: it is developed from the lateral bud of the sub taro.

During the growth and development of mother taro, from the perspective of component content, the weight gradually increased, and the water content gradually decreased. In addition, the total starch content gradually increased, and the soluble sugar content initially increased and then decreased. From a cytological point of view, the volume of parenchymal cells continued to increase, and the internal starch content gradually increased. The number and diameter of starch bodies showed an upward trend, and the number of starch granules in a single amyloplast also increased (*Zhu et al., 2017*). During taro bulb expansion, the number of annual rings on the surface of the bulb gradually increased. In addition, the volume of parenchymal cells in the bulb continued to increase. Moreover, the number and size of amyloplasts gradually increased; thus, the entire cell was enriched; with the expansion of the bulb, the vascular tissue gradually developed and perfected, and the area of the sieve tube increased in number and became irregularly arranged; the taro bulb was densely covered with mucous cavities, which spread out from the center of the bulb, but no mucus cavity was observed in the epidermal cells (*Sheng, 2021*).

**Amyloplast enrichment of taro bulbs**

At present, the research on the development and proliferation of amyloplasts primarily focuses on endosperm cells of grains. In the development of endosperm amyloplasts, the number of plastids continued to divide, and the number increased; then, the number and volume of starch granules increased in the plastid; individual starch granules increased in size and then began to form small amyloplasts. As the development progressed, the envelope of small amyloplasts was further extended and expanded. Finally, the newly expanded envelope was filled with starch granules, thereby forming large amyloplasts. The amyloplasts of large starch granules in wheat endosperm formed small starch granules by budding, whereas the amyloplasts of small starch granules proliferated by constriction or budding (*Wei, Lan & Xu, 2002*; *Wei et al., 2008*).

A few reports have focused on the proliferation mode of amyloplasts in underground rhizome crops. The starch bodies of the complex starch contained in sweet potato tubers exist in the form of "dyads," "triad," and "polyad." For the growth mode of the amyloplast,

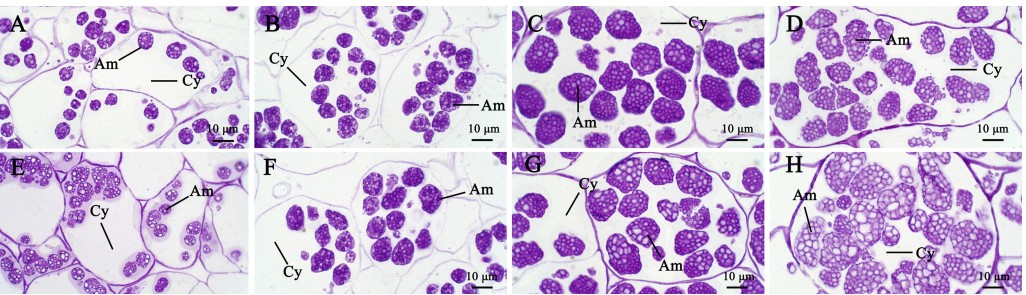

**Figure 4** **The development process of amyloplasts in the parenchyma cells of Kui taro bulbs.** (A,E) Bulb diameter 1 cm; (B,F) bulb diameter five cm; (C,G) bulb diameter nine cm; (D,H) bulb diameter 13 cm; (Am) amyloid; (Cy) cytoplasm.

single-grain starch directly expands and grows; the amyloplasts of multigranular starch initially split into monomers and then grow in the form of monomers; the amyloplasts of complex starch do not divide during growth and form large complex amyloplasts (*Jing et al., 2013*). Based on the morphology of amyloids observed by scanning electron microscopy, the cassava root amyloid membrane is either "constricted" or "wrinkled" or "sprouted," forming multiple differentiation centers of amyloplasts; with the continuous expansion of the amyloplast, the membrane structure was degraded and disappeared, and irregular starch granules were released (*Min et al., 2010*).

In general, the accumulation of taro bulb starch is divided into three stages, namely, starch formation, rapid starch accumulation, and amyloplast enrichment. The specific performance is the formation of amyloplasts and the continuous increase in diameter, followed by the continuous increase in the number of starches, and finally the increase in the number of starches in amyloplasts. The surface of taro amyloplast is mostly round and oval, belonging to complex starch, containing multiple polyhedral starch granules (*e.g.*, Take Kui taro). In the early stage of development, amyloplast was mostly distributed at the edge of the cell and then gathered at the center of the cell. At later developmental stages, the number and size of amyloplasts continued to increase (Fig. 4). Transmission electron microscopy showed that the amyloplasts of taro bulbs were large, and large amyloplasts split into several small starch granules. Some small starch granules (Fig. 5A) were close to the free state at the edge of amyloplasts, and these granules were loosely arranged and smaller in size. With the development of the bulb, starch granules with large diameters in the amyloplasts were mostly concentrated in one area, and their arrangement was relatively compact. Small starch granules were mostly distributed on the other side or around, and their arrangement was loose (Fig. 5B).

Taro bulb starch originates from the precursor plastid, and starch granules are formed in the plastid and free in the cytoplasm of the membrane (*Sheng, 2021*). With the increase of starch granules, the volume of plastids increases, and starch granules develop from having an irregular shape to round. In the later stage, they gradually extruded one another into polygons. The amyloplast of taro bulbs was mostly complex starches. Amyloid proliferation is divided into two stages. In the first stage, the number of starch granules in the amyloplast

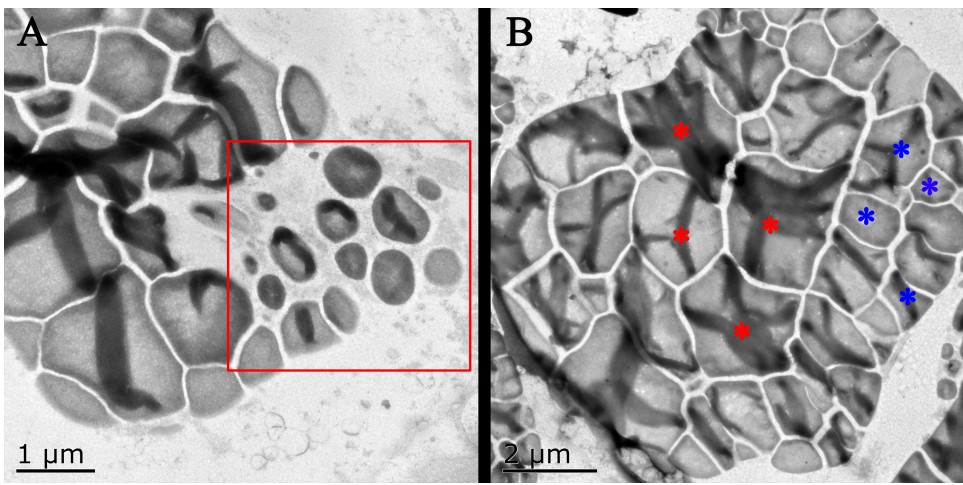

**Figure 5** **Transmission electron microscope pictures of starch grains of Kui taro bulbs.** (A) Bulb diameter 1 cm; The proliferation of amyloplasts, the red box shows the small starch granules after the overflow and shrinkage, and the shape has changed from irregular to spherical. (B) Bulb diameter 5 cm; The arrangement of small starch granules in large starch bodies, red asterisks represent large starch granules, blue asterisks represent small starch granules, two types of starch granules are distributed on both sides of starch bodies.

changes; the starch granules gradually increase, fill up, squeeze, and deform one another. Large starch granules split into many small starch granules. The second stage involves the proliferation of amyloplasts. Taro amyloplasts can be split *via* membrane constriction proliferation and capsular vesicle proliferation. During the constriction and proliferation of envelope, the starch granules in the amyloplast are squeezed to both sides through the inward depression of the amyloplast envelope. Furthermore, it is constricted into multiple amyloplasts, and the encapsulated vesicles proliferate in which the original amyloplasts spit out vesicles from the envelope, and new amyloplasts are generated and proliferated in the vesicles.

## Effects of environmental and cultivation methods on the development of taro bulbs

Taro adapts to high-temperature and high-humidity environments, and it is not resistant to low temperature and frost. The optimum temperature for germination is 12 °C–15 °C. In general, the optimum temperature for growth is 25 °C–30 °C. If the temperature is low, the growth of taro will slow down or stop. If the temperature is high, the condition will not be conducive to the development and expansion of taro bulb (*Wang, Wang & Lu, 2007*). Different varieties of taro have different requirements for temperature. Multi-cormel taro can be planted at low temperatures, making it widely distributed in temperate zones. Kui taro has strict requirements for high temperature. Therefore, Kui taro is mostly produced in tropical and subtropical regions with high temperature and humidity (*Chang & Wang, 2019*). Taro requires sunlight, and the light saturation point is approximately 50,000 lx. Taro is shade tolerant, and it can grow under scattered light. The light intensity, composition, and light time remarkably affect the growth of taro, but strong light is conducive to the

growth of taro, which improves yield and quality. Under blue-violet light, the leaves of taro are large and thick, and the petioles are thick and short; such conditions are conducive to the growth and development of bulbs. Under red and yellow light, the leaves are small, and the petioles are slender; thus, such conditions are not conducive to the growth and development of bulbs. In the early stage of taro development, a longer light time is necessary to increase the leaf area and the accumulation of photosynthetic products. The later stage of taro development requires a shorter light time to facilitate the formation and expansion of bulbs (*Chang & Wang, 2019*). Taro requires dampness, but differences are observed among classifications. Aquatic taro is grown in paddy fields, but dry taro cannot be flooded for a long time. Taro has different requirements for humidity before different growth stages. The field should be kept moist during the germination stage to induce the germination of taro. During the plant vigorous growth stage, the water demand is large, and water supply must be ensured; the soil should be kept dry before harvesting to maintain a good condition for the harvesting and storage of bulbs (*Huang, Ke & Sun, 2016*). Taro does not require strict soil texture. Loose and fertile soil with a deep soil layer as well as convenient irrigation and drainage are conducive to the growth of taro and expansion of bulbs. Taro can grow normally in soil with pH of 4.1–9.1, but the optimum pH is 5.5–7.0. A highly acidic or highly alkaline soil is not conducive to the growth and development of taro (*Huang, Ke & Sun, 2016*).

Different cultivation methods remarkably affect the yield of taro. Film mulching can provide soil temperature in the early stage of taro growth, which promotes the growth of taro and the expansion of bulbs. The growth and yield of taro bulbs in perforated film-covering cultivation was better than that in ridge film-covering cultivation. However, considering the inconvenience of cultivating a large amount of soil during the growth period of taro, ridge and perforated film-covering cultivation easily form green taro, thereby affecting the quality and taste of bulbs (*Wang et al., 2001*). If no freezing damage is observed after emergence, then early sowing is conducive to the development of taro root. This condition will lead to an increase in plant height, size, and yield, but these changes only slightly affect the number of taro and the shape index of taro. If planting is late, then the life cycle of the taro will be shortened, which is not conducive to the growth and development of the bulb. This condition will lead to insufficient bulb expansion, thereby reducing the yield (*Zheng, 2008*). Nitrogen and potassium fertilizers have evident effects on the yield and quality of taro, and potassium fertilizer has a greater effect than nitrogen fertilizer. A remarkable interaction effect was observed among nitrogen, potassium, and phosphorus fertilizer. Within the reasonable range of potassium and nitrogen fertilizer application, the yield gradually increases with the increase of the amount of fertilizer. Excessive fertilizer application will reduce the yield. Phosphate fertilizer alone only slightly affects the development of taro bulbs, but no evident rule has been established. Furthermore, reasonable fertilization promotes the growth, development, and yield increase of taro bulbs (*Song & Xu, 2004*).

## Regulation of hormones on bulb development
### *Regulation of endogenous hormones on bulb expansion and starch enrichment*

Hormones are important endogenous substances that regulate plant growth, and it is a key factor in bulb formation (*Durbak, Yao & McSteen, 2012*). Some genes and proteins related to bulb formation are also closely related to plant hormone signaling pathways (*Aksenova et al., 2012*). Different plant hormones have different functions in bulb expansion, and gibberellin (GA) can inhibit or delay tuber formation (*Vreugdenhil & Struik, 1989*). Abscisic acid (ABA) does not participate in the induced metamorphosis of tubers, but it counteracts the antagonism of other hormones (*Shu, Zhou & Yang, 2017*). Auxin (IAA) can promote the metamorphic development of tubers and plant root development, and its concentration affects tuberous root thickening (*Wang et al., 2006*). Although GA and ABA are not directly related to tuber formation, they are related to the ratio of GA3/ABA. The balance of "inducing substances" and "inhibiting substances" is a key factor for tuber formation (*Liu, 2001*). Cytokinins (CTK) are primarily involved in the formation of tubers. *Matsuo & Mitsuzono (1988)* reported that the content of zeatin riboside (ZR) is significantly and positively correlated with the formation and thickening of sweet potato tubers (*Matsuo & Mitsuzono, 1988*). The overexpression of CTK-synthesized gene ipt in potato could form more tubers (*Tao et al., 2010*). In particular, IAA-related genes such as auxin response factor and Aux/IAAs are expressed during the early tuber development (*Kloosterman et al., 2008*). IAA and GA3 are necessary for potato stolon elongation. ABA and jasmonic acid (JA) are positive regulators for inducing tuber formation. GA3 is a negative regulator (*Liu et al., 2019*). JA and methyl jasmonate, as classes of plant growth regulators, play an important role in tuber and bulb formation (*Sarkar, Pandey & Sharma, 2006*).

The regulation of plant endogenous hormones is closely related to starch anabolism (*Kim & Kim, 2005*). The enlargement of plant bulbs is primarily dependent on starch accumulation and cell division enlargement, and starch accumulation is primarily dependent on sucrose synthesis and transportation. Plant hormone signal transduction affects starch accumulation. ABA can induce the expression of starch synthesis genes and enhance the transduction of sugar signals to promote starch accumulation (*Akihiro, Mizuno & Fujimura, 2005*). The level of GA at the grain filling stage of wheat is positively correlated with the final grain yield and starch yield. GA plays an important role in starch accumulation in wheat grains. Changes in endogenous hormone levels may indirectly affect starch accumulation in grains by affecting regulatory enzymes and regulatory processes (*Xie et al., 2003*). Scientists added IAA to the MS medium, and the potato tuber starch content and starch granule size increased by 15%–30% (*Gukasyan et al., 2005*). In the study of tulip bulbs, IAA and ZR indirectly promoted starch accumulation by increasing the activity of ADP-glucose pyrophosphorylase (AGPase), thereby catalyzing the production of a large number of products. Endogenous hormones may promote starch accumulation by participating in the starch synthesis pathway (*Miao et al., 2016*). Hormones have multiple roles, and they interact to form a regulatory network, thereby regulating tuber development (*Jung & McCouch, 2013*). In the early stage of taro development (bulb is approximately one cm in diameter), the content of endogenous hormones (ABA), bulb endogenous

zeatin (Z), and ZR showed an upward trend, whereas the content of IAA, GA3, and JA showed a downward trend. In the later stage of development (bulb is approximately 13 cm in diameter), the content of endogenous hormones (ABA, IAA, Z, and GA3) showed an upward trend. The content of ZR and JA showed a downward trend, but the content of IAA, GA3, and JA was generally high during the whole development process (*Sheng, 2021*). Other related studies on the effect of endogenous hormones on the growth and development of taro bulbs and starch enrichment have not been conducted.

### Regulation of the growth, development, and starch enrichment of bulbs by exogenous hormones

The effect of exogenous hormones on the rhizome expansion of potato, sweet potato, and other tuberous crops has been studied. *Yang (2005)* used four auxins to spray potatoes (*Yang, 2005*). The results showed that the four auxins increased plant height and stem diameter as well as prolonged the photosynthetic accumulation in the later stage. This condition allowed the tubers to accumulate more organic matter during expansion, thereby substantially increasing the yield. The exogenous application of IAA can promote the formation of potato stolons and the development of tubers. It is achieved by accelerating starch accumulation and starch granule enlargement, which promote the formation and development of tubers (*Gukasyan et al., 2005*; *Roumeliotis et al., 2012a*; *Roumeliotis, Visser & Bachem, 2012b*). GA also promotes the occurrence of stolons. Stolons appeared on the second day after the medium containing GA3 and IAA was added, and the occurrence continued throughout the tuber setting period (*Lian et al., 2002*). However, the addition of GA could inhibit or delay the formation of potato tubers, and the inactivation of the active GA gene could promote potato tuber formation (*Xu et al., 1998*; *Roumeliotis et al., 2012a*; *Roumeliotis, Visser & Bachem, 2012b*). Treatment with exogenous GAs inhibited sucrose synthase (SS) and soluble starch synthase (SSS) activity, thereby decreasing the content of sucrose and starch in tubers (*Vreugdenhil & Sergeeva, 1999*). ABA is a factor promoting the formation of potato tubers, and timely spraying is beneficial to potato formation (*Krauss & Marschner, 1982*; *Garcia, Stritzler & Capiati, 2014*). Varying results about the role of ABA in the development of tuber plants have been obtained. GA3 inhibits the formation of potatoes *in vitro*, whereas ABA promotes tuber formation (*Hu et al., 2017*). Exogenously applied ABA can promote tuber expansion (*Xu et al., 2022*). However, *in vitro*, ABA cannot make stolons metamorphose into tubers smoothly (*Yang, 2005*). ABA does not participate in the induced metamorphosis of tubers, but its presence counteracts the respective physiological activity of other hormones (*Xu et al., 2022*). CTK can promote potato tuber development, regulate the balance between source and sink, and participate in the transport of nutrients to storage organs (*Roitsch & Ehneß, 2000*). When a certain concentration of CTK is applied exogenously, the biomass of tubers remarkably increases, and the transformation of stolons to tubers is accelerated (*Romanov, 2009*). *In vitro*, CTK inhibits sucrose invertase activity but activates phosphorylase and AGPase, thereby promoting starch accumulation (*Zhu, Luo & Fan, 2016*). Therefore, CTK is an important factor inducing tuber formation (*Quan, Zhang & Cao, 2002*). The exogenous application of JA and its derivatives can induce the swelling of the stolon top, and the content of endogenous JA increases during this process

(*Abdala et al., 2002*). After exogenous JA treatment, intracellular sucrose accumulates, thereby increasing the osmotic pressure of the cell wall, changing the structure of the cell wall, and increasing cell ductility. In addition, more polysaccharides such as cellulose, hemicellulose, and pectin are accumulated, indicating that JA controls the expansion of the cell by regulating the synthesis of intracellular sugar (*Takahashi et al., 1995*). This phenomenon induces the formation of the apical meristem of potato stolon and promotes tuber development (*Cenzano et al., 2003*).

The development of taro bulbs is remarkably affected by exogenous hormones. The diameter of taro bulbs that were irrigated with auxin increased remarkably; the weight increased, and the filling degree of amyloplasts in parenchymal cells increased. The low concentration of 6-BA can promote the development of bulbs, but it is not conducive to the enrichment of amyloplasts. High concentrations of 6-BA have a certain inhibitory effect on the development of bulbs. GA3 promotes the elongation of the petioles of taro plants, but it does not promote the expansion of the bulbs. High concentrations of GA3 (100–200 mg/L) have an inhibitory effect on the development of taro bulbs, but it promotes the development of taro and increases the number of taro (*Sheng, 2021*). Limited studies have focused on the effects of exogenous hormones on the development of taro bulbs and the enrichment of amyloplasts; thus, further research must be conducted.

## Role of key enzyme genes in starch synthesis during starch enrichment

In crops primarily underground storage organs, the synthesis and accumulation of starch are complex physiological and biochemical processes, which result from the synergistic interaction of multiple enzymes. The key enzymes of starch synthesis in root crops, such as potato and lotus root, have been widely studied. The changes in AGPase and SSS activities have important effects on starch synthesis in potato tubers (*Tang, 2015*). However, SS and AGPase can remarkably promote the synthesis of starch during lotus root rhizome expansion, and their activities affect the starch content of lotus root rhizomes at the mature stage (*Li et al., 2006*). Based on the study of substance accumulation and changes in related enzyme activities during the development of yam, sucrose phosphate synthase activity plays a key regulatory role in the development of yam tubers, and it is closely related to the main functional substances (*Liang et al., 2011*). Based on the study of taro bulbs, AGPase activity is positively correlated with total starch content (*Zang et al., 2016*). With the gradual deepening of research on starch metabolism pathways, people have obtained novel insights into the key enzyme gene sequences and related expression regulators in the pathway.

In sweet potato, the key enzyme genes of starch synthesis such as AGPase and SS have been cloned into the gene sequence. The expression and regulation of these genes have been studied, and these key enzyme genes play a key role in the sweet potato starch metabolism pathway (*Tang et al., 2011*). The genes controlling sweet potato starch synthesis include granule-bound starch synthase (GBSS) gene I, SSS genes I and II, starch-branching enzyme (SBE) genes I and II, starch-debranching enzyme gene, AGPase gene A/B/C, SS genes I and II, and isoamylase gene (*Kim et al., 2009*).

GBSS I is a key enzyme that controls starch synthesis, and it catalyzes the synthesis of amylose. *Otani et al. (2007)* interfered with the expression of GBSS I by RNAi technology to make sweet potato taste more glutinous. SSSII can affect the structure of amylopectin and reduce the gelatinization temperature of starch (*Otani et al., 2007*). The reduction of starch gelatinization temperature is conducive to simplifying starch hydrolysis and reducing the production cost of starch fermentation (*Takahata et al., 2010*). AGPase improves the starch content of potato tubers (*Song, Xie & Liu, 2005*). However, the synergistic expression of starch synthesis-related genes under exogenous sucrose treatment promotes the conversion of sucrose to starch (*Ahn et al., 2010*). Peak synthase has been widely studied, but no direct research has been conducted on taro starch synthase.

## CONCLUSIONS AND FUTURE DIRECTION

The development of taro bulbs is a complex biochemical process, including the accumulation of morphogenesis and assimilation products, involving gene expression, material metabolism, nutrient input, and the effect of external environmental conditions (Fig. 6). However, limited studies have been conducted on the development of taro bulbs worldwide. Understanding the development process, expansion mechanism, and regulation mechanism of taro bulbs plays an important role in the production of taro as well as in ensuring food security and responding to food crises. At present, research on the development of taro primarily focuses on evolutionary classification, genotype and isozyme analysis, cultivation, production, processing, and utilization. Limited research has been conducted on the physiological mechanism and hormone regulatory pathway of taro growth and development, taro bulb expansion, key gene expression, and starch enrichment. Therefore, the following research will become the key research direction in the future.

### Expansion of taro bulbs and regulation of the development and spatial distribution of starch bodies

Starch is the main storage material of taro bulbs, and amyloplasts are organelles that synthesize and accumulate starch. The development of amyloplasts determines the yield and quality of taro. Limited studies locally and abroad have focused on the development of taro corm and amyloplast, and they remain in the preliminary stage. The observation on the fine structure of amyloplast and its proliferation mode as well as the spatial distribution characteristics of taro corm amyloplast remain unclear. In the future, the occurrence, division, proliferation, and enrichment of amyloplasts in parenchymal cells of different types of taro bulbs; the differences in the physical and chemical properties of taro starch at different developmental stages; and the development and enrichment characteristics of amyloplasts in different spatial parts should be focused on.

### Role of key enzyme genes in starch synthesis and enrichment

Starch is the main storage material of taro bulbs. The expansion of taro is closely related to the synthesis of starch, and genes related to starch synthesis are closely related to starch enrichment, which directly determine the starch content of taro. Research on starch

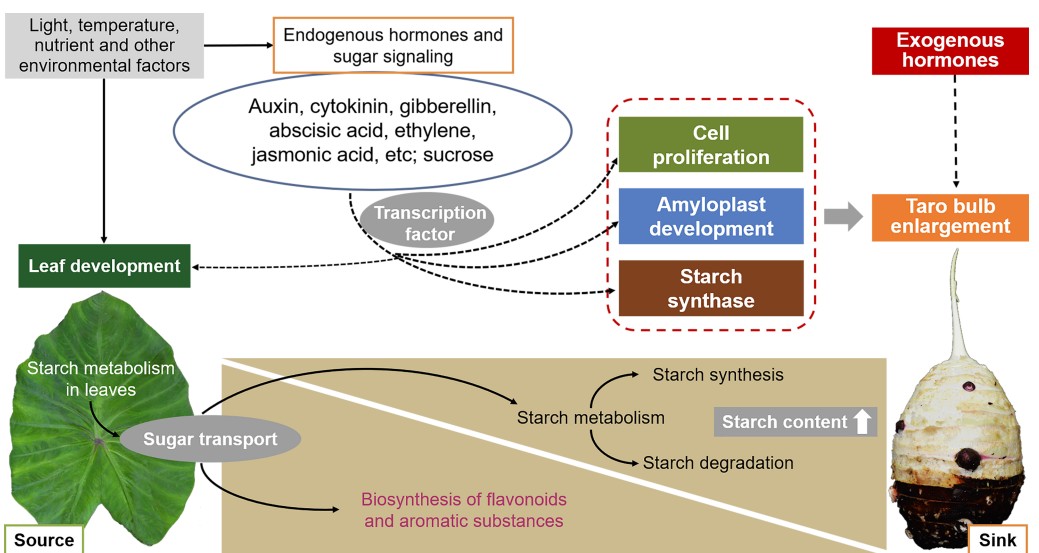

**Figure 6** Effects of external environmental factors and internal factors on taro bulb expansion and starch enrichment.

synthase gene has remarkably progressed in wheat, rice, potato, and other crops, but limited research has been conducted on taro starch synthase gene. Therefore, the differences in the expression of key enzyme genes (AGPase, GBSS, SSS, and SBE) for taro starch synthesis, the role of these genes in regulating starch enrichment, and the exploration of individual gene functions will become the focus of research.

## Regulation of hormones on bulb swelling and starch enrichment

The expansion of taro bulbs primarily depends on the increase in the number and volume of parenchymal cells, and this process results from the synergistic action of various hormones, particularly IAA, GA, and CTK. Changes in hormones during bulb development and the relationship between hormones and bulb expansion must be investigated. Exploring the relationship between the expression of hormone synthesis genes and signal transduction-related genes and the enrichment of taro starch is of great research significance.

## Hormone-regulated pathways promoting bulb expansion and starch enrichment

Exogenous hormones and plant growth regulators have important regulatory effects on taro bulb swelling, starch enrichment, and yield increase. Therefore, the effects of exogenous substances, such as 6-BA, 2,4-D, GA3, PP333, and 5-aminolevulinic acid, on the development, yield, and quality of taro bulb, as well as the type and concentration of the best exogenous hormone to promote corm expansion and starch enrichment must be studied. Moreover, plant growth regulators must be developed to improve the quality of taro. All efforts will provide important theoretical basis for taro production

The development of taro bulbs still requires considerable research. With the deepening of research and the solution to key problems, the production of taro will continue to improve, and the development and utilization of taro will be more efficient.

### Funding
The authors received no funding for this work.

### Competing Interests
The authors declare there are no competing interests.

### Author Contributions
- Erjin Zhang conceived and designed the experiments, performed the experiments, analyzed the data, prepared figures and/or tables, authored or reviewed drafts of the article, and approved the final draft.
- Wenyuan Shen conceived and designed the experiments, performed the experiments, analyzed the data, prepared figures and/or tables, authored or reviewed drafts of the article, and approved the final draft.
- Weijie Jiang performed the experiments, prepared figures and/or tables, authored or reviewed drafts of the article, and approved the final draft.
- Wenlong Li performed the experiments, prepared figures and/or tables, authored or reviewed drafts of the article, and approved the final draft.
- Xiaping Wan performed the experiments, prepared figures and/or tables, authored or reviewed drafts of the article, and approved the final draft.
- Xurun Yu analyzed the data, prepared figures and/or tables, authored or reviewed drafts of the article, and approved the final draft.
- Fei Xiong conceived and designed the experiments, authored or reviewed drafts of the article, and approved the final draft.

### Data Availability
    This is a literature review.

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
