# Peer review of "Research progress on the bulb expansion and starch enrichment in taro (Colocasia esculenta (L). Schott)"

_PeerJ, doi:10.7717/peerj.15400_

## Round 0.1 · original submission · Major Revisions

Please revise your manuscript according to the comments from the reviewers.

Reviewer 1 ·

Basic reporting

According to the instruction of the review, I will summarize the manuscript point by point.
(1) The author summarized the bulb expansion and starch enrichment in taro, which is underground bulbous crop. Researchers focus on crops (maize, rice, wheat, potato, yam, etc ) will be the potential readers of this review article. The end user such as food manufacturer will also have potential interest. Hence, it is of broad and cross-disciplinary interest.
(2) I don't find a review focus on taro bulb expansion yet, hence, it hasn't been reviewed recently.
(3) Basically, in the introduction part, the author introduced the subject clearly, however, there are some knowledge need to be added or re-organized. Here are:
Line 39-41: The author mentioned the origin, cultivated country and the history of taro. But, have all sites or countries cultivated taro more than 2000 years?
Line 43-45: The author described different classification criteria of taro. However, in line 45, we don’t know what kind of classification of the “The three classifications of taro……”
Line 52-53, The author described chromosome numbers of Kui taro, and multi-head taro, but didn’t describe the multi-cormels taro, please add the progresses of it.
Line 56-73:Please organize this section in in-depended paragraph. In addition, the logic of this part needs to be reorganized. I suggest the author describe all the nutrient components firstly, then describe their utilization according to nutrient components or the structure of starch. To avoid repeat, the distribution of the taro could be deleted.

Experimental design

no comment

Validity of the findings

no comment

Additional comments

Although the title or sub-titles are logically arranged. The manuscript still needs to be improved before be accepted for publication. Firstly, I suggest the author improved the language by colleagues or a professional editor. Many of the sentences lack necessary links. For example, line 132-139. Other questions are as follows:
(1) In the part of the "Tara expansion and development process", I suggested author summarize the growth curve of taro as a whole, then the development of mother-taro, sub-taro and their growth. Due to the discrimination of the three types of taros, the author should point out whether all of them have same axillary-bud growth pattern? Furthermore, the abbreviation of the gene or protein name in the text should be in full name as their first appearance.
(2) The part of ‘Amyloplast enrichment process of taro bulbs’
Line 141-160. These two paragraphs summarized the results of other plants not the taro. To understand the research gap among taro and other plant, I suggest author compare research progress to other plant, not only summarize the results of other plant.
Line 161-163, the two papers cited in the text didn’t describe the accumulation process of taro bulb.
In line 177, Does taro bulb starch originate from the precursor plastid? I’m not sure. But I thought that starch gradually accumulate in the precursor plastid is more reasonable.
Line 211, “Calla taro……, but dry taro……”. These two kinds of taro belong to what type of taro described in Figure 2?
(3) The part of ‘Regulation of hormones on bulb development’.
In this part, most of the progresses are from other plants but not taro, I suggest the author summarize the progresses of the taro firstly, then turn to the knowledge of other plant, pointed the research gap among taro and other plants. Furthermore, Line 279-283, summarized two stages of the development, but not mention what kind of plants. If it is the taro, the author didn ‘t describes the criteria of the different development stages of taro.
Line 279 and 281, “Z” is what kind of hormone?
Line 296, “In the present experiment……”, this should not be included in a review paper.
(4) Part of “Role of key enzyme genes in starch synthesis in starch enrichment”
This part has similar question as the part mentioned above. Progresses of the taro are very limited. Authors should re-organize this part.
In addition, a few typeset mistakes are highlighted on the PDF version of the manuscript.

Annotated reviews are not available for download in order to protect the identity of reviewers who chose to remain anonymous.

Reviewer 2 ·

Basic reporting

The manuscript is well-written and easy to read. The accompanying figures are of reasonable quality.

This manuscript is definitely of interest for those working in this field. While there are numerous reviews on taro but mostly focused on the nutritional aspects whereas this manuscript emphasized on the biochemistry and physiology of the growth and development of taro, i.e., bulb swelling and starch enrichment. Such information can be applied into future research to improve the production of taro.

Suggestions for improvement:
There are two versions of abstracts in the manuscript file. Please select the one that conforms to the format of PeerJ.

I noticed that there are certain parts/paragraphs that lack citation. For instance, lines 193-200. Please check throughout the manuscript.

Some additional details could be added to the figure legends for the benefit of the readers.
Figure 1-3: Please name the researcher who authenticate the species and indicate the scientific name in each figure legend. The source of the samples - cultivated in the lab or purchased?
Figure 3: Suggest to provide brief (one sentence) description of the various labelled structures
Figure 4-5: Did the authors prepare these microscopic images specifically this manuscript? Otherwise, please cite the source of the images.
Figure 4: Suggest to include the photographs of the bulbs of different sizes.

Experimental design

The methodology used in the collection of articles are clearly described. Some of the references used are based on other tubes such as potatoes where the authors draw comparison with taro. I feel that this could be highlighted in the Materials and Methods section too.

Some suggestions:

For the subsection on effects of environmental and cultivation methods on the development of taro and related topics, it is suggested that the findings by different researchers to be tabulated (instead of included in the text in the current manuscript). Reason being, it will be easy for the potential readers to get clearer picture on the development in this field. The authors can provide an overall summary for each subsection, highlighting the achievements and gaps - this will facilitate further research.

While there are extensive details on the anatomy of the plant, and the physiology of the taro expansion process, the biochemistry aspects of the changes that accompany taro expansion (apart from starch enrichment) and studies on various external and internal factors, seems not well-elaborated. The authors should consider this.

Validity of the findings

Overall, the authors managed to compiled relevant information and summarised them systematically in the present manuscript. One way the impact of this manuscript can be enhanced is to provide more critical analysis and interpretation of the findings that the authors have compiled and summarised.

Some suggestions:

Lines 95-190: The authors can insert a schematic diagram that shows how the growth and development (expansion) of taro.

The authors can summarise the tools that have been used to study the growth, development and morphogenesis of taro, and what are the other tools that, perhaps, can be utilised to gain more insights into this aspect.

The authors are suggested to include their perspectives on how an improved understanding of this aspect may have practical applications in the agriculture sector and relevant industries.

Additional comments

Overall this manuscript represents a compilation of information that are reflects the progress on our research to understand the growth, development and morphogenesis of taro, and the opportunities for further research. The above comments and suggestions should be considered to enhance the quality of the review.

Reviewer 3 ·

Basic reporting

Taro is a crop for food, vegetables and feed, and its bulb is rich in starch. The starch granules are small and easy to digest, resulting in unique dietary and health care functions of taro. The enlargement and starch enrichment of corm determine the yield and quality of taro, and they are regulated by plant hormones. However, few information is investigated about the mechanism of corm expansion and the law of starch filling at home and abroad. This article reviewed the formation and development of taro bulb at the cytological level and the changes in bulb expansion and starch enrichment at physiological levels especially involving endogenous hormones and key enzyme genes for starch synthesis. The future research directions and research focus about the development of taro bulb were also prospected. The research progress on the bulb expansion and starch enrichment in taro is relatively systematically and comprehensively. The review work can not only deepen the research on underground storage organ development mechanism of tuber crops such as taro and potato, but also have important significance for excavating the potential of taro production, improving the quality of taro, and the industrial development of taro starch. Furthermore, the most interesting work present some good and clear images taken by the authors.
I have some comments and would recommend the work for publication in Peer J after revision.

Comments:
1.Background and Conclusions parts needed to be reorganized due to some repeated contents.
2.The Abstract Parts should be concise and specific.
3. Line 336, Role of key enzyme genes in starch synthesis in starch enrich-ment should be changed to Role of key enzyme genes in starch synthesis during starch enrichment
4. Line 368, Conclusions and future direction should be described respectively
5.Fig.1A/B/C/D/E/F should be described respectively in the manuscript
6.Fig.6 note: taro bulb swelling change to bulb expand, which is consistent with the title.
7. The format of references should be proofread according to the Peer J requirement.
8.I suggested that the manuscript should be polished by a professional English speaker or commercial English language editing services before publication.

Experimental design

good

Validity of the findings

good

---

## Round 0.2 · Minor Revisions

Please make revision according to the comments from the reviewers.

Reviewer 1 ·

Basic reporting

No comments

Experimental design

no comment

Validity of the findings

Some parts content of the manuscript is not well fit the subtitle, I pointed out them in the first-round review, please author considering.

Additional comments

Generally, the manuscript has improved after the revision, however, to increase the readability of the article, it still needs improve before accept for publishing. Here are some suggestions to polish the article.
1. Authors need to consider the logic of the part of line 94-124, especially, authors showed point out the purpose to describe the CPSs protein and TC1 gene from line 108 to 118.
2. In the part of the “Amyloplast enrichment process of taro bulbs”, authors cited many researches in other plants, however, the similarities and differences among taro and other plants have not been summarized, hence, this part lack internal connection. The same problem occurs in the “Regulation of hormones on bulb development” and “Roles of key enzyme genes in starch synthesis during starch enrichment”. Please authors consider.
3. Typeset mistakes, such as, starch-es in line 163, com-plex in line 180, gran-ules in line 186, etc. Please authors check the whole manuscript carefully.
4. The author's name in line 116 is inconsistent with references.
5. Line 214, the taro-forming stage, was not described in the Figure 2.
6. Line 344, “SS” here, refers to “soluble starch synthase” or the “sucrose synthase”, was not very clear.
7. I suggest author deleted “locally and abroad” in line 374 and 387, since “Peer J” is an international magazine and readers may be worldwide.
8. The author only shows the dimeter of the bulb on Figure 2, however, the author divided life circle of taro growth into five stages in Line 97-99. What are the relationships between these five stages and the size of bulb?
9. On figure 6, the accumulation of substance in taro includes starch and flavonoids, and aromatic substances. However, in the text, the biosynthesis of flavonoids and aromatic substances were not included. Authors may need to interpret this.

Reviewer 2 ·

Basic reporting

The authors have revised the manuscript based on the comments from the reviewers, however, there are some issues that still require clarification.

1. The person who authenticated the species (Colocasia esculenta) grown in the authors' lab should be named, and the manner by which the authentication was done (based on morphology?) should be stated.

2. Suggest to include the age (weeks/months) (in the form of range) of the different developmental stages.

3. Please clarify if terms like "mother taro", "sub taro", and "grandson taro" are the commonly accepted botanical terms or just translation of local Chinese common name. Suggest to use the correct scientific/botanical terms in the manuscript to avoid confusion.

4. Previous studies should be compiled in the form of a table rather than to list them in the text. In the current form, there is a mixture of literature on taro and other related crops, and it is not exactly clear if the conclusions were drawn from previous studies on only taro or other crops.

Experimental design

No comment

Validity of the findings

No comment

Additional comments

Refer above

Reviewer 3 ·

Basic reporting

The revised manuscript has much improved. All previous questions and suggestions have been addressed.

Experimental design

well

Validity of the findings

well

Additional comments

The revised manuscript has much improved. All previous questions and suggestions have been addressed.

---

## Round 0.3 · Minor Revisions

Please make correction accordingly.

Reviewer 1 ·

Basic reporting

No comment.

Experimental design

No comment

Validity of the findings

No comment

Additional comments

Please author check the former comments.

Reviewer 2 ·

Basic reporting

The authors misunderstood my earlier query on the authentication of plant materials used in the laboratory. Please include in the figure legend the name of the botanist who identified the plants as Colocasia esculenta and whether it was achieved by morphological examination or molecular methods.

I feel that proper botanical terms should be used, however, when that's impossible, care should be taken to ensure that the readers are aware that some of the terminologies used are direct translation of certain local languages and are not considered as the official terminologies for that subject matter.

Lines 99-103: the authors should clarify the terms that they used "mother taro", "grandson taro", "great grandson taro" are merely direct translation of the local Chinese terms

Figure 3: same as above

Figure 2: the authors did not include the rough estimation of the age (in days/week) for each developmental stage

Experimental design

No comment

Validity of the findings

No comment

---

## Round 0.4 · Minor Revisions

The reviewer has a single remaining concern. Please revise accordingly.

Reviewer 2 ·

Basic reporting

The authors have responded to my previous queries. Regarding my comment on identification of plant sample, I was referring specifically to those used in Figures 1-3 (presumably from the authors' laboratory). I have suggested the name of the botanist who identified the authors' samples, NOT the person who gave the scientific name to the species, to be included in the figure legends. I feel this is important as the samples shown in the figures lack morphological characteristics that may aid in identification.
Thank you.

Experimental design

No comment

Validity of the findings

No comment

---

## Round 0.5 · Minor Revisions

The information was added. It is almost ready to be accepted for publication.

The Section Editor noted that the manuscript needs to be edited for grammar throughout:

> Please review and improve the writing. There are many grammar errors in the text. For example:
"Taro is an important potato crop in the world, which can be used as food, vegetable, feed and industrial raw materials.": remove 'in the world' to clarify the relative clause; before 'and,' a comma is missing.
"it can be classified into two classification". etc. etc.

---

## Round 0.6 · Minor Revisions

Robert Winkler, the Section Editor, has commented and said:

"The quality of the writing is still insufficient for publication. I uploaded the manuscript to Grammarly, and it marked >200 errors!"

Please provide evidence for professional proofreading before resubmission.

---

## Round 0.7 · accepted · Accept

Language editing was done.